# VIDEOFLOW: A CONDITIONAL FLOW-BASED MODEL FOR STOCHASTIC VIDEO GENERATION

**Manoj Kumar**[*]**, Mohammad Babaeizadeh, Dumitru Erhan,**
**Chelsea Finn, Sergey Levine, Laurent Dinh, Durk Kingma**
Google Research, Brain Team
{mechcoder,mbz,dumitru,chelseaf,slevine,laurentdinh,durk}@google.com

## ABSTRACT

Generative models that can model and predict sequences of future events can, in principle, learn to capture complex real-world phenomena, such as physical interactions. However, a central challenge in video prediction is that the future is highly uncertain: a sequence of past observations of events can imply many possible futures. Although a number of recent works have studied probabilistic models that can represent uncertain futures, such models are either extremely expensive computationally as in the case of pixel-level autoregressive models, or do not directly optimize the likelihood of the data. To our knowledge, our work is the first to propose multi-frame video prediction with normalizing flows, which allows for direct optimization of the data likelihood, and produces high-quality stochastic predictions. We describe an approach for modeling the latent space dynamics, and demonstrate that flow-based generative models offer a viable and competitive approach to generative modeling of video.

## 1 INTRODUCTION

Exponential progress in the capabilities of computational hardware, paired with a relentless effort towards greater insights and better methods, has pushed the field of machine learning from relative obscurity into the mainstream. Progress in the field has translated to improvements in various capabilities, such as classification of images (Krizhevsky et al., 2012), machine translation (Vaswani et al., 2017) and super-human game-playing agents (Mnih et al., 2013; Silver et al., 2017), among others. However, the application of machine learning technology has been largely constrained to situations where large amounts of supervision is available, such as in image classification or machine translation, or where highly accurate simulations of the environment are available to the learning agent, such as in game-playing agents. An appealing alternative to supervised learning is to utilize large unlabeled datasets, combined with predictive generative models. In order for a complex generative model to be able to effectively predict future events, it must build up an internal representation of the world. For example, a predictive generative model that can predict future frames in a video would need to model complex real-world phenomena, such as physical interactions. This provides an appealing mechanism for building models that have a rich understanding of the physical world, without any labeled examples. Videos of real-world interactions are plentiful and readily available, and a large generative model can be trained on large unlabeled datasets containing many video sequences, thereby learning about a wide range of real-world phenoma. Such a model could be useful for learning representations for further downstream tasks (Mathieu et al., 2016), or could even be used directly in applications where predicting the future enables effective decision making and control, such as robotics (Finn et al., 2016). A central challenge in video prediction is that the future is highly uncertain: a short sequence of observations of the present can imply many possible futures. Although a number of recent works have studied probabilistic models that can represent uncertain futures, such models are either extremely expensive computationally (as in the case of pixel-level autoregressive models), or do not directly optimize the likelihood of the data.

In this paper, we study the problem of stochastic prediction, focusing specifically on the case of conditional video prediction: synthesizing raw RGB video frames conditioned on a short context

---

[*]A majority of this work was done as part of the Google AI Residency Program.

of past observations (Ranzato et al., 2014; Srivastava et al., 2015; Vondrick et al., 2015; Xingjian et al., 2015; Boots et al., 2014). Specifically, we propose a new class of video prediction models that can provide exact likelihoods, generate diverse stochastic futures, and accurately synthesize realistic and high-quality video frames. The main idea behind our approach is to extend flow-based generative models (Dinh et al., 2014; 2016) into the setting of conditional video prediction. To our knowledge, flow-based models have been applied only to generation of non-temporal data, such as images (Kingma & Dhariwal, 2018), and to audio sequences (Prenger et al., 2018). Conditional generation of videos presents its own unique challenges: the high dimensionality of video sequences makes them difficult to model as individual datapoints. Instead, we learn a latent dynamical system model that predicts future values of the flow model's latent state. This induces Markovian dynamics on the latent state of the system, replacing the standard unconditional prior distribution. We further describe a practically applicable architecture for flow-based video prediction models, inspired by the Glow model for image generation (Kingma & Dhariwal, 2018), which we call VideoFlow.

Our empirical results show that VideoFlow achieves results that are competitive with the state-of-the-art in stochastic video prediction on the action-free BAIR dataset, with quantitative results that rival the best VAE-based models. VideoFlow also produces excellent qualitative results, and avoids many of the common artifacts of models that use pixel-level mean-squared-error for training (e.g., blurry predictions), without the challenges associated with training adversarial models. Compared to models based on pixel-level autoregressive prediction, VideoFlow achieves substantially faster test-time image synthesis [1], making it much more practical for applications that require real-time prediction, such as robotic control (Finn & Levine, 2017). Finally, since VideoFlow directly optimizes the likelihood of training videos, without relying on a variational lower bound, we can evaluate its performance directly in terms of likelihood values.

## 2 RELATED WORK

Early work on prediction of future video frames focused on deterministic predictive models (Ranzato et al., 2014; Srivastava et al., 2015; Vondrick et al., 2015; Xingjian et al., 2015; Boots et al., 2014). Much of this research on deterministic models focused on architectural changes, such as predicting high-level structure (Villegas et al., 2017b), energy-based models (Xie et al., 2017), generative cooperative nets (Xie et al., 2020), ABPTT (Xie et al., 2019), incorporating pixel transformations (Finn et al., 2016; De Brabandere et al., 2016; Liu et al., 2017) and predictive coding architectures (Lotter et al., 2017), as well as different generation objectives (Mathieu et al., 2016; Vondrick & Torralba, 2017; Walker et al., 2015) and disentangling representations (Villegas et al., 2017a; Denton & Birodkar, 2017). With models that can successfully model many deterministic environments, the next key challenge is to address stochastic environments by building models that can effectively reason over uncertain futures. Real-world videos are always somewhat stochastic, either due to events that are inherently random, or events that are caused by unobserved or partially observable factors, such as off-screen events, humans and animals with unknown intentions, and objects with unknown physical properties. In such cases, since deterministic models can only generate one future, these models either disregard potential futures or produce blurry predictions that are the superposition or averages of possible futures.

A variety of methods have sought to overcome this challenge by incorporating stochasticity, via three types of approaches: models based on variational auto-encoders (VAEs) (Kingma & Welling, 2013; Rezende et al., 2014), generative adversarial networks (Goodfellow et al., 2014), and autoregressive models (Hochreiter & Schmidhuber, 1997; Graves, 2013; van den Oord et al., 2016b;c; Van Den Oord et al., 2016).

Among these models, techniques based on variational autoencoders which optimize an evidence lower bound on the log-likelihood have been explored most widely (Babaeizadeh et al., 2017; Denton & Fergus, 2018; Lee et al., 2018; Xue et al., 2016; Li et al., 2018). To our knowledge, the only prior class of video prediction models that directly maximize the log-likelihood of the data are autoregressive models (Hochreiter & Schmidhuber, 1997; Graves, 2013; van den Oord et al., 2016b;c; Van Den Oord et al., 2016), that generate the video one pixel at a time (Kalchbrenner et al., 2017). However, synthesis with such models is typically inherently sequential, making synthesis substantially

---

[1]We generate 64x64 videos of 20 frames in less than 3.5 seconds on a NVIDIA P100 GPU as compared to the fastest autoregressive model for video (Reed et al., 2017) that generates a frame every 3 seconds

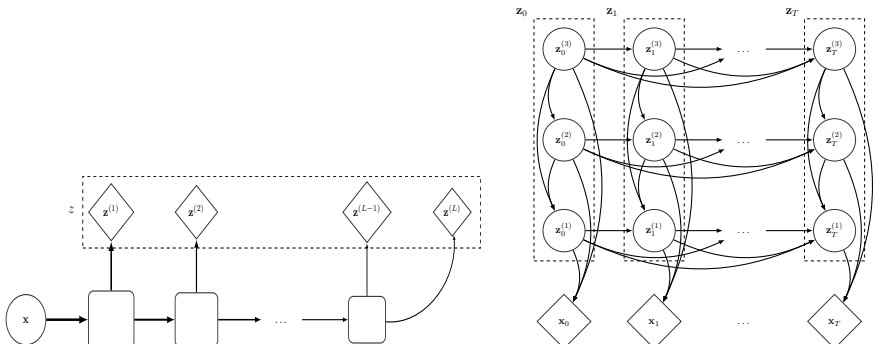

**Figure 1: Left: Multi-scale prior** The flow model uses a multi-scale architecture using several levels of stochastic variables. **Right: Autoregressive latent-dynamic prior** The input at each timestep $\mathbf{x}_t$ is encoded into multiple levels of stochastic variables $(\mathbf{z}_t^{(1)}, \ldots, \mathbf{z}_t^{(L)})$. We model those levels through a sequential process $\prod_t \prod_l p(\mathbf{z}_t^{(l)} \mid \mathbf{z}_{<t}^{(l)}, \mathbf{z}_t^{(>l)})$.

inefficient on modern parallel hardware. Prior work has aimed to speed up training and synthesis with such auto-regressive models (Reed et al., 2017; Ramachandran et al., 2017). However, (Babaeizadeh et al., 2017) show that the predictions from these models are sharp but noisy and that the proposed VAE model produces substantially better predictions, especially for longer horizons. In contrast to autoregressive models, we find that our proposed method exhibits faster sampling, while still directly optimizing the log-likelihood and producing high-quality long-term predictions.

## 3 Preliminaries: Flow-Based Generative Models

*Flow-based generative models* (Dinh et al., 2014; 2016) have a unique set of advantages: exact latent-variable inference, exact log-likelihood evaluation, and parallel sampling. In *flow-based generative models* (Dinh et al., 2014; 2016), we infer the latent variable $\mathbf{z}$ corresponding to a datapoint $\mathbf{x}$, by transforming $\mathbf{x}$ through a composition of invertible functions $\mathbf{f} = \mathbf{f}_1 \circ \mathbf{f}_2 \circ \cdots \circ \mathbf{f}_K$. We assume a tractable prior $p_{\boldsymbol{\theta}}(\mathbf{z})$ over latent variable $\mathbf{z}$, for eg. a Logistic or a Gaussian distribution. By constraining the transformations to be invertible, we can compute the log-likelihood of $\mathbf{x}$ exactly using the *change of variables* rule. Formally,

$$\log p_{\boldsymbol{\theta}}(\mathbf{x}) = \log p_{\boldsymbol{\theta}}(\mathbf{z}) + \sum_{i=1}^{K} \log |\det(d\mathbf{h}_i/d\mathbf{h}_{i-1})| \tag{1}$$

where $\mathbf{h}_0 = \mathbf{x}$, $\mathbf{h}_i = \mathbf{f}_i(\mathbf{h}_{i-1})$, $\mathbf{h}_K = \mathbf{z}$ and $|\det(d\mathbf{h}_i/d\mathbf{h}_{i-1}|$ is the Jacobian determinant when $\mathbf{h}_{i-1}$ is transformed to $\mathbf{h}_i$ by $\mathbf{f}_i$. We learn the parameters of $\mathbf{f}_1 \ldots \mathbf{f}_K$ by maximizing the log-likelihood, i.e Equation (1), over a training set. Given $\mathbf{g} = \mathbf{f}^{-1}$, we can now generate a sample $\hat{\mathbf{x}}$ from the data distribution, by sampling $\mathbf{z} \sim p_{\boldsymbol{\theta}}(\mathbf{z})$ and computing $\hat{\mathbf{x}} = \mathbf{g}(\mathbf{z})$.

## 4 Proposed Architecture

We propose a generative flow for video, using the standard multi-scale flow architecture in (Dinh et al., 2016; Kingma & Dhariwal, 2018) as a building block. In our model, we break up the latent space $\mathbf{z}$ into separate latent variables per timestep: $\mathbf{z} = \{\mathbf{z}_t\}_{t=1}^{T}$. The latent variable $\mathbf{z}_t$ at timestep $t$ is an invertible transformation of a corresponding frame of video: $\mathbf{x}_t = \mathbf{g}_{\boldsymbol{\theta}}(\mathbf{z}_t)$. Furthermore, like in (Dinh et al., 2016; Kingma & Dhariwal, 2018), we use a multi-scale architecture for $\mathbf{g}_{\boldsymbol{\theta}}(\mathbf{z}_t)$ (Fig. 1): the latent variable $\mathbf{z}_t$ is composed of a stack of multiple levels: where each level $l$ encodes information about frame $\mathbf{x}_t$ at a particular scale: $\mathbf{z}_t = \{\mathbf{z}_t^{(l)}\}_{l=1}^{L}$, one component $\mathbf{z}_t^{(l)}$ per level.

## 4.1 Invertible multi-scale architecture

We first briefly describe the invertible transformations used in the multi-scale architecture to infer $\{\mathbf{z}_t^{(l)}\}_{l=1}^L = \mathbf{f}_{\boldsymbol{\theta}}(\mathbf{x}_t)$ and refer to (Dinh et al., 2016; Kingma & Dhariwal, 2018) for more details. For convenience, we omit the subscript $t$ in this subsection. We choose invertible transformations whose Jacobian determinant in Equation 1 is simple to compute, that is a triangular matrix, diagonal matrix or a permutation matrix as explored in prior work (Rezende & Mohamed, 2015; Deco & Brauer, 1995). For permutation matrices, the Jacobian determinant is one and for triangular and diagonal Jacobian matrices, the determinant is simply the product of diagonal terms.

- Actnorm: We apply a learnable per-channel scale and shift with data-dependent initialization.
- Coupling: We split the input $y$ equally across channels to obtain $y_1$ and $y_2$. We compute $z_2 = f(y_1) * y_2 + g(y_1)$ where $f$ and $g$ are deep networks. We concat $y_1$ and $z_2$ across channels.
- SoftPermute: We apply a 1x1 convolution that preserves the number of channels.
- Squeeze: We reshape the input from $H \times W \times C$ to $H/2 \times W/2 \times 4C$ which allows the flow to operate on a larger receptive field.

We infer the latent variable $z^{(l)}$ at level $l$ using:

$$\text{Flow}(y) = \text{Coupling}(\text{SoftPermute}(\text{Actnorm}(y)))) \times N \tag{2}$$
$$\text{Flow}_l(y) = \text{Split}(\text{Flow}(\text{Squeeze}(y))) \tag{3}$$
$$(\mathbf{h}^{(>l)}, \mathbf{z}^l) \leftarrow \text{Flow}_l(\mathbf{h}^{(>l-1)}) \tag{4}$$

where $N$ is the number of steps of flow. In Equation (3), via Split, we split the output of Flow equally across channels into $\mathbf{h}^{(>l)}$, the input to $\text{Flow}_{(l+1)}(.)$ and $z^{(l)}$, the latent variable at level $l$. We, thus enable the flows at higher levels to operate on a lower number of dimensions and larger scales. When $l = 1$, $\mathbf{h}^{(>l-1)}$ is just the input frame $x$ and for $l = L$ we omit the Split operation. Finally, our multi-scale architecture $\mathbf{f}_{\boldsymbol{\theta}}(\mathbf{x}_t)$ is a composition of the flows at multiple levels from $l = 1 \ldots L$ from which we obtain our latent variables i.e $\{\mathbf{z}_t^{(l)}\}_{l=1}^L$.

## 4.2 Autoregressive latent dynamics model

We use the multi-scale architecture described above to infer the set of corresponding latent variables for each individual frame of the video: $\{\mathbf{z}_t^{(l)}\}_{l=1}^L = \mathbf{f}_{\boldsymbol{\theta}}(\mathbf{x}_t)$; see Figure 1 for an illustration. As in Equation (1), we need to choose a form of latent prior $p_{\boldsymbol{\theta}}(\mathbf{z})$. We use the following autoregressive factorization for the latent prior:

$$p_{\boldsymbol{\theta}}(\mathbf{z}) = \prod_{t=1}^T p_{\boldsymbol{\theta}}(\mathbf{z}_t | \mathbf{z}_{<t}) \tag{5}$$

where $\mathbf{z}_{<t}$ denotes the latent variables of frames prior to the $t$-th timestep: $\{\mathbf{z}_1, ..., \mathbf{z}_{t-1}\}$. We specify the conditional prior $p_{\boldsymbol{\theta}}(\mathbf{z}_t | \mathbf{z}_{<t})$ as having the following factorization:

$$p_{\boldsymbol{\theta}}(\mathbf{z}_t | \mathbf{z}_{<t}) = \prod_{l=1}^L p_{\boldsymbol{\theta}}(\mathbf{z}_t^{(l)} | \mathbf{z}_{<t}^{(l)}, \mathbf{z}_t^{(>l)}) \tag{6}$$

where $\mathbf{z}_{<t}^{(l)}$ is the set of latent variables at previous timesteps and at the same level $l$, while $\mathbf{z}_t^{(>l)}$ is the set of latent variables at the same timestep and at higher levels. See Figure 1 for a graphical illustration of the dependencies.

We let each $p_{\boldsymbol{\theta}}(\mathbf{z}_t^{(l)} | \mathbf{z}_{<t}^{(l)}, \mathbf{z}_t^{(>l)})$ be a conditionally factorized Gaussian density:

$$p_{\boldsymbol{\theta}}(\mathbf{z}_t^{(l)} | \mathbf{z}_{<t}^{(l)}, \mathbf{z}_t^{(>l)}) = \mathcal{N}(\mathbf{z}_t^{(l)}; \boldsymbol{\mu}, \sigma) \tag{7}$$
$$\text{where } (\boldsymbol{\mu}, \log \sigma) = NN_{\boldsymbol{\theta}}(\mathbf{z}_{<t}^{(l)}, \mathbf{z}_t^{(>l)}) \tag{8}$$

| Model | Fooling rate |
|---------|--------------|
| SAVP-VAE | 16.4 % |
| VideoFlow | **31.8 %** |
| SV2P | 17.5 % |

**Table 1:** We compare the realism of the generated trajectories using a real-vs-fake 2AFC Amazon Mechanical Turk with SAVP-VAE and SV2P.



**Figure 2:** We condition the VideoFlow model with the frame at t = 1 and display generated trajectories at t = 2 and t = 3 for three different shapes.

where $NN_{\boldsymbol{\theta}}(.)$ is a deep 3-D residual network (He et al., 2015) augmented with dilations and gated activation units and modified to predict the mean and log-scale. We describe the architecture and our ablations of the architecture in Section D and E of the appendix.

In summary, the log-likelihood objective of Equation (1) has two parts. The invertible multi-scale architecture contributes $\sum_{i=1}^{K} \log |\det(d\mathbf{h}_i/d\mathbf{h}_{i-1})|$ via the sum of the log Jacobian determinants of the invertible transformations mapping the video $\{\mathbf{x}_t\}_{t=1}^{T}$ to $\{\mathbf{z}_t\}_{t=1}^{T}$; the latent dynamics model contributes $\log p_{\boldsymbol{\theta}}(\mathbf{z})$, i.e Equation (5). We jointly learn the parameters of the multi-scale architecture and latent dynamics model by maximizing this objective.

Note that in our architecture we have chosen to let the prior $p_{\boldsymbol{\theta}}(\mathbf{z})$, as described in eq. (5), model temporal dependencies in the data, while constraining the flow $\mathbf{g}_{\boldsymbol{\theta}}$ to act on separate frames of video. We have experimented with using 3-D convolutional flows, but found this to be computationally overly expensive compared to an autoregressive prior; in terms of both number of operations and number of parameters. Further, due to memory limits, we found it only feasible to perform SGD with a small number of sequential frames per gradient step. In case of 3-D convolutions, this would make the temporal dimension considerably smaller during training than during synthesis; this would change the model's input distribution between training and synthesis, which often leads to various temporal artifacts. Using 2-D convolutions in our flow $\mathbf{f}_{\boldsymbol{\theta}}$ with autoregressive priors, allows us to synthesize arbitrarily long sequences without introducing such artifacts.

## 5 EXPERIMENTS

All our generated videos and qualitative results can be viewed at this website. In the generated videos, a border of blue represents the conditioning frame, while a border of red represents the generated frames.

### 5.1 VIDEO MODELLING WITH THE STOCHASTIC MOVEMENT DATASET

We use VideoFlow to model the Stochastic Movement Dataset used in (Babaeizadeh et al., 2017). The first frame of every video consists of a shape placed near the center of a 64x64x3 resolution gray background with its type, size and color randomly sampled. The shape then randomly moves in one of eight directions with constant speed. (Babaeizadeh et al., 2017) show that conditioned on the first frame, a deterministic model averages out all eight possible directions in pixel space. Since the shape moves with a uniform speed, we should be able to model the position of the shape at the $(t+1)^{th}$ step using only the position of the shape at the $t^{th}$ step. Using this insight, we extract random temporal patches of 2 frames from each video of 3 frames. We then use VideoFlow to maximize the log-likelihood of the second frame given the first, i.e the model looks back at just one frame. We observe that the bits-per-pixel on the holdout set reduces to a very low $0.04$ bits-per-pixel for this model. On generating videos conditioned on the first frame, we observe that the model consistently predicts the future trajectory of the shape to be one of the eight random directions. We compare our model with two state-of-the-art stochastic video generation models SV2P and SAVP-VAE (Babaeizadeh et al., 2017; Lee et al., 2018) using their Tensor2Tensor implementation (Vaswani et al., 2018). We assess the quality of the generated videos using a real vs fake Amazon Mechanical Turk test. In the test, we inform the rater that a "real" trajectory is one in which the shape is consistent in color and congruent

| Model | Bits-per-pixel |
|-------|----------------|
| VideoFlow | **1.87** |
| SAVP-VAE | $\leq 6.73$ |
| SV2P | $\leq 6.78$ |

**Table 2: Left:** We report the average bits-per-pixel across 10 target frames with 3 conditioning frames for the BAIR action-free dataset.

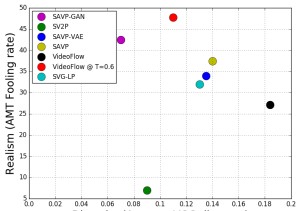

**Figure 3:** We measure realism using a 2AFC test and diversity using mean pairwise cosine distance between generated samples in VGG perceptual space.

throughout the video. We show that VideoFlow outperforms the baselines in terms of fooling rate in Table 1 consistently generating plausible "real" trajectories at a greater rate.

## 5.2 VIDEO MODELING WITH THE BAIR DATASET

We use the action-free version of the BAIR robot pushing dataset (Ebert et al., 2017) that contain videos of a Sawyer robotic arm with resolution 64x64. In the absence of actions, the task of video generation is completely unsupervised with multiple plausible trajectories due to the partial observability of the environment and stochasticity of the robot actions. We train the baseline models, SAVP-VAE, SV2P and SVG-LP to generate 10 target frames, conditioned on 3 input frames. We extract random temporal patches of 4 frames, and train VideoFlow to maximize the log-likelihood of the 4th frame given a context of 3 past frames. We, thus ensure that all models have seen a total of 13 frames during training.

**Bits-per-pixel**: We estimated the variational bound of the bits-per-pixel on the test set, via importance sampling, from the posteriors for the SAVP-VAE and SV2P models. We find that VideoFlow outperforms these models on bits-per-pixel and report these values in Table 2. We attribute the high values of bits-per-pixel of the baselines to their optimization objective. They do not optimize the variational bound on the log-likelihood directly due to the presence of a $\beta \neq 1$ term in their objective and scheduled sampling (Bengio et al., 2015).

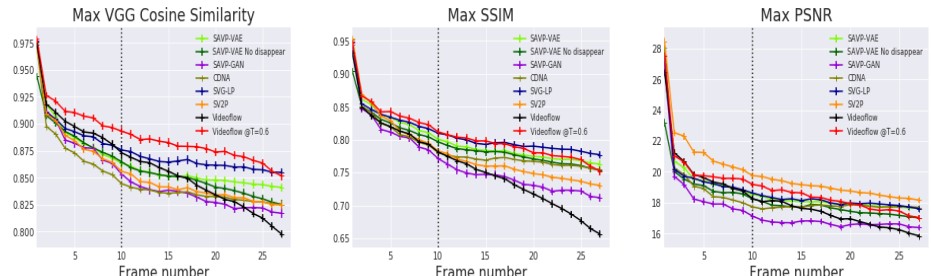

**Figure 4:** For a given set of conditioning frames on the BAIR action-free we sample 100 videos from each of the stochastic video generation models. We choose the video closest to the ground-truth on the basis of PSNR, SSIM and VGG perceptual metrics and report the best possible value for each of these metrics. All the models were trained using ten target frames but are tested to generate 27 frames. For all the reported metrics, **higher is better**.

**Accuracy of the best sample**: The BAIR robot-pushing dataset is highly stochastic and the number of plausible futures are high. Each generated video can be super realistic, can represent a plausible future in theory but can be far from the single ground truth video perceptually. To partially overcome this, we follow the metrics proposed in prior work (Babaeizadeh et al., 2017; Lee et al., 2018; Denton & Fergus, 2018) to evaluate our model. For a given set of conditioning frames in the BAIR action-free test-set, we generate 100 videos from each of the stochastic models. We then compute the closest of these generated videos to the ground truth according to three different metrics, PSNR (Peak Signal to

Noise Ratio), SSIM (Structural Similarity) (Wang et al., 2004) and cosine similarity using features obtained from a pretrained VGG network (Dosovitskiy & Brox, 2016; Johnson et al., 2016) and report our findings in Figure 4. This metric helps us understand if the true future lies in the set of all plausible futures according to the video model.

In prior work, (Lee et al., 2018; Babaeizadeh et al., 2017; Denton & Fergus, 2018) effectively tune the pixel-level variance as a hyperparameter and sample from a deterministic decoder. They obtain training stabiltiy and improve sample quality by removing pixel-level noise using this procedure. We can remove pixel-level noise in our VideoFlow model resulting in higher quality videos at the cost of diversity by sampling videos at a lower temperature, analogous to the procedure in (Kingma & Dhariwal, 2018). For a network trained with additive coupling layers, we can sample the $t^{th}$ frame $x_t$ from $P(x_t|x_{<t})$ with a temperature $T$ simply by scaling the standard deviation of the latent gaussian distribution $P(z_t|z_{<t})$ by a factor of $T$. We report results with both a temperature of 1.0 and the optimal temperature tuned on the validation set using VGG similarity metrics in Figure 4. Additionally, we also applied low-temperature sampling to the latent gaussian priors of SV2P and SAVP-VAE and empirically found it to hurt performance. We report these results in Figure 12

For SAVP-VAE, we notice that the hyperparameters that perform the best on these metrics are the ones that have disappearing arms. For completeness, we report these numbers as well as the numbers for the best performing SAVP models that do not have disappearing arms. Our model with optimal temperature performs better or as well as the SAVP-VAE and SVG-LP models on the VGG-based similarity metrics, which correlate well with human perception (Zhang et al., 2018) and SSIM. Our model with temperature $T = 1.0$ is also competent with state-of-the-art video generation models on these metrics. PSNR is explicitly a pixel-level metric, which the VAE models incorporate as part of its optimization objective. VideoFlow on the other-hand models the conditional probability of the joint distribution of frames, hence as expected it underperforms on PSNR.

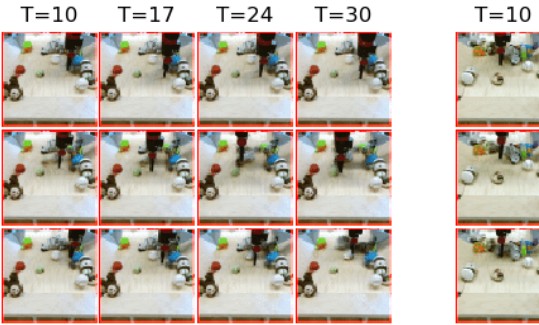
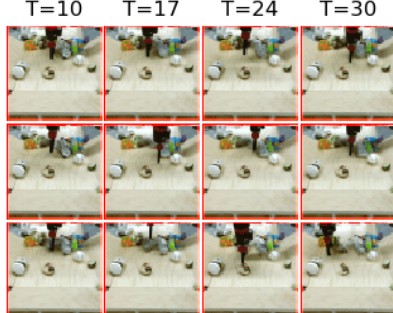

**Figure 5:** We display three different futures for two sets of conditioning frames (left and right) at $T = 0.6$ showcasing diversity in outcomes

**Diversity and quality in generated samples:** For each set of conditioning frames in the test set, we generate 10 videos and compute the mean distance in VGG perceptual space across these 45 different pairs. We average this across the test-set for $T = 1.0$ and $T = 0.6$ and report these numbers in Figure 3. We also assess the quality of the generated videos at $T = 1.0$ and $T = 0.6$, using a real vs fake Amazon Mechanical Turk test and report fooling rates. We observe that VideoFlow outperforms diversity values reported in prior work (Lee et al., 2018) while being competitive in the realism axis. We also find that VideoFlow at $T = 0.6$ has the highest fooling rate while being competent with state-of-the-art VAE models in diversity.

On inspection of the generated videos, we find that at lower temperatures, the arm exhibits less random behaviour with the background objects remaining static and clear achieving higher realism scores. At higher temperatures, the motion of arm is much more stochastic, achieving high diversity scores with the background objects becoming much noisier leading to a drop in realism.

**Fréchet Video Distance (FVD):** We evaluate VideoFlow using the recently proposed Fréchet Video Distance (FVD) metric (Unterthiner et al., 2018), an adaptation of the Fréchet Inception Distance (FID) metric (Heusel et al., 2017) for video generation. (Unterthiner et al., 2018) report results with models trained on a total of 16 frames with 2 conditioning frames; while we train our VideoFlow

| # Frames Seen: Training | | | | |
|---|---|---|---|---|
| Conditioning | 3 | 3 | 3 | 2 |
| Total | 13 | 13 | 13 | 16 |
| # Frames: Evaluation | | | | |
| Ground truth | 3 | 3 | 2 | 2 |
| Total | 13 | 16 | 16 | 16 |
| **Model** | **FVD** | | | |
| VideoFlow (T=0.8) | 95±4 | 127±3 | 131±5 | - |
| VideoFlow (T=1.0) | 149±6 | 221±8 | 251±7 | - |
| SAVP | - | - | - | 116 |
| SV2P | - | - | - | 263 |

Table 3: **Fréchet Video Distance:**. We report the mean and standard deviation across 5 runs for 3 different frame settings. Results are not directly comparable across models due to the differences between the total number of frames seen during training and the number of conditioning frames.

model on a total of 13 frames with 3 conditioning frames, making our results not directly comparable to theirs. We evaluate FVD for both shorter and longer rollouts in Table 3. We show that, even in the settings that are disadvantageous to VideoFlow, where we compute the FVD on a total of 16 frames, when trained on just 13 frames, VideoFlow performs comparable to SAVP.

## 5.3 LATENT SPACE INTERPOLATION

**BAIR robot pushing dataset**: We encode the first input frame and the last target frame into the latent space using our trained VideoFlow encoder and perform interpolations. We find that the motion of the arm is interpolated in a temporally cohesive fashion between the initial and final position. Further, we use the multi-level latent representation to interpolate representations at a particular level while keeping the representations at other levels fixed. We find that the bottom level interpolates the motion of background objects which are at a smaller scale while the top level interpolates the arm motion.

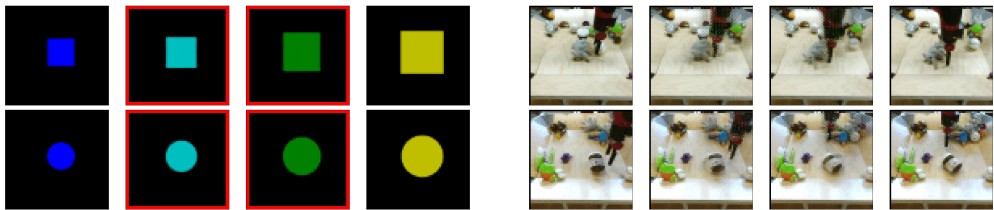

Figure 6: **Left:** We display interpolations between a) a small blue rectangle and a large yellow rectangle b) a small blue circle and a large yellow circle. **Right:** We display interpolations between the first input frame and the last target frame of two test videos in the BAIR robot pushing dataset.

**Stochastic Movement Dataset:** We encode two different shapes with their type fixed but a different size and color into the latent space. We observe that the size of the shape gets smoothly interpolated. During training, we sample the colors of the shapes from a uniform discrete distribution which is reflected in our experiments. We observe that all the colors in the interpolated space lie in the set of colors in the training set.

## 5.4 LONGER PREDICTIONS

We generate 100 frames into the future using our model trained on 13 frames with a temperature of 0.5 and display our results in Figure 7. On the top, even 100 frames into the future, the generated frames remain in the image manifold maintaining temporal consistency. In the presence of occlusions,

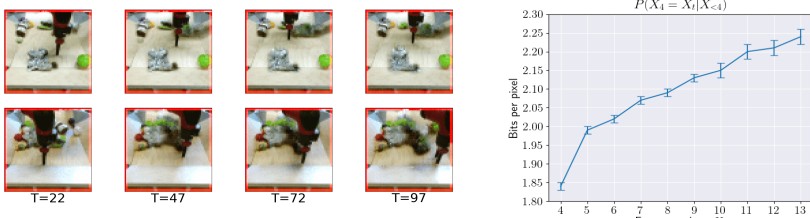

**Figure 7: Left:** We generate 100 frames into the future with a temperature of 0.5. The top and bottom row correspond to generated videos in the absence and presence of occlusions respectively. **Right:** We use VideoFlow to detect the plausibility of a temporally inconsistent frame to occur in the immediate future.

the arm remains super-sharp but the background objects become noisier and blurrier. Our VideoFlow model has a bijection between the $\mathbf{z}_t$ and $\mathbf{x}_t$ meaning that the latent state $\mathbf{z}_t$ cannot store information other than that present in the frame $\mathbf{x}_t$. This, in combination with the Markovian assumption in our latent dynamics means that the model can forget objects if they have been occluded for a few frames. In future work, we would address this by incorporating longer memory in our VideoFlow model; for example by parameterizing $NN_{\boldsymbol{\theta}}()$ as a recurrent neural network in our autoregressive prior (eq. 8) or using more memory-efficient backpropagation algorithms for invertible neural networks (Gomez et al., 2017).

### 5.5 Out-of-sequence detection

We use our trained VideoFlow model, conditioned on 3 frames as explained in Section 5.2, to detect the plausibility of a temporally inconsistent frame to occur in the immediate future. We condition the model on the first three frames of a test-set video $X_{<4}$ to obtain a distribution $P(X_4|X_{<4})$ over its 4th frame $X_4$. We then compute the likelihood of the $t^{\text{th}}$ frame $X_t$ of the same video to occur as the 4th time-step using this distribution. i.e, $\mathcal{P}(X_4 = X_t|X_{<4})$ for $t = 4 \ldots 13$. We average the corresponding bits-per-pixel values across the test set and report our findings in Figure 7. We find that our model assigns a monotonically decreasing log-likelihood to frames that are more far out in the future and hence less likely to occur in the 4th time-step.

## 6 Open source code and checkpoints

We open-source the implementation of our code in the Tensor2Tensor codebase. We additionally open-source various components of our trained VideoFlow model, to evaluate log-likelihood, to generate frames and compute latent codes as reusable TFHub modules

## 7 Conclusion and Discussion

We describe a practically applicable architecture for flow-based video prediction models, inspired by the Glow model for image generation Kingma & Dhariwal (2018), which we call VideoFlow. We introduce a latent dynamical system model that predicts future values of the flow model's latent state replacing the standard unconditional prior distribution. Our empirical results show that VideoFlow achieves results that are competitive with the state-of-the-art VAE models in stochastic video prediction. Finally, our model optimizes log-likelihood directly making it easy to evaluate while achieving faster synthesis compared to pixel-level autoregressive video models, making our model suitable for practical purposes. In future work, we plan to incorporate memory in VideoFlow to model arbitrary long-range dependencies and apply the model to challenging downstream tasks.

ACKNOWLEDGEMENTS

We would like to thank Ryan Sepassi and Lukasz Kaiser for their extensive help in using Tensor2Tensor, Oscar Täckström for finding a bug in our evaluation pipeline that improved results across all models, Ruben Villegas for providing code for the SVG-LP baseline and Mostafa Dehghani for providing feedback on a draft of the rebuttal.

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

## A    MOVING MNIST - QUALITATIVE EXPERIMENTS

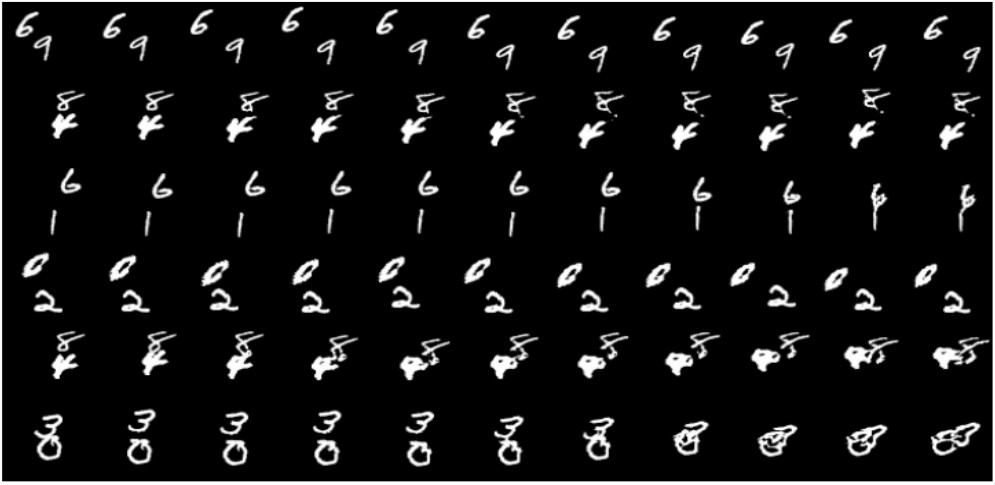

**Figure 8:** We display ten frame rollouts conditioned on a single frame on the Moving MNIST dataset.

Similar to the Stochastic Movement Dataset as described in Section 5.1, we extract random temporal patches of 2 frames on the Moving MNIST dataset (Srivastava et al., 2015). We train our VideoFlow model to maximize the log-likelihood of the second frame, given the first. Our rollouts over 10 frames capture realistic digit movement.

## B    HUMAN3.6M - QUALITATIVE EXPERIMENTS

We model the Human3.6M dataset (Ionescu et al., 2014), by maximizing the log-likelihood of the 4th frame given the first three frames, in a random temporal patch of 4 frames. We observe that on this dataset, our model fails to capture reasonable human motion. We hope that by increasing model capacity and using more expressive priors, we can acheive better performance on this dataset in the future.

## C    DISCRETIZATION AND UNIFORM QUANTIZATION

Let $\mathcal{D} = \{\mathbf{x}^{(i)}\}_{i=1}^{N}$ be our dataset of i.i.d. observations of a random variable $\mathbf{x}$ with an unknown true distribution $p^*(\mathbf{x})$. Our data consist of 8-bit videos, with each dimension rescaled to the domain $[0, 255/256]$. We add a small amount of uniform noise to the data, $\mathbf{u} \sim \mathcal{U}(0, 1/256.)$, matching its discretization level (Dinh et al., 2016; Kingma & Dhariwal, 2018). Let $q(\mathbf{x})$ be the resulting empirical distribution corresponding to this scaling and addition of noise. Note that additive noise is required to prevent $q(\mathbf{x})$ from having infinite densities at the datapoints, which can result in ill-behaved optimization of the log-likelihood; it also allows us to recast maximization of the log-likelihood as minimization of a KL divergence.

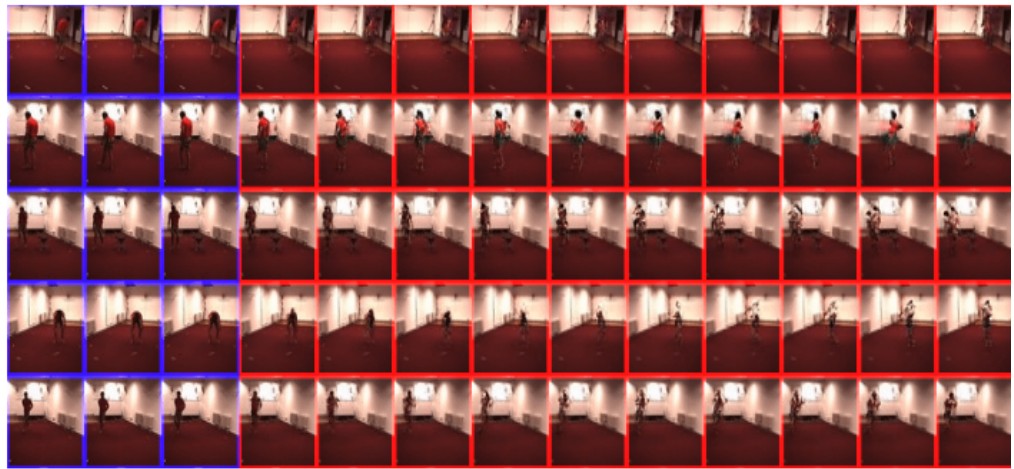

**Figure 9:** We display ten frame rollouts conditioned on 3 frames on the Human3.6M dataset.

## D  RESIDUAL NETWORK ARCHITECTURE

Here we'll describe the architecture for the residual network $NN_{\boldsymbol{\theta}}()$ that maps $\mathbf{z}^{(l)}_{<t}, \mathbf{z}^{(>l)}_t$ to $(\boldsymbol{\mu}^{(l)}_t, \log \sigma^{(l)}_t)$ (Left: Figure 10). As shown in the left of Figure 10, let $\mathbf{h}^{(>l)}_t$ be the tensor representing $\mathbf{z}^{(>l)}_t$ after the split operation between levels in the multi-scale architecture. We apply a $1 \times 1$ convolution over $\mathbf{h}^{(>l)}_t$ and concatenate this across channels to each latent from the previous time-step and the same-level independently. In this way, we obtain $((W\mathbf{h}^{(>l)}_t; \mathbf{z}^{(l)}_{t-1}), (W\mathbf{h}^{(>l)}_t; \mathbf{z}^{(l)}_{t-2}) \ldots (W\mathbf{h}^{(>l)}_t; \mathbf{z}^{(l)}_{t-n}))$. We transform these values into $(\boldsymbol{\mu}^{(l)}_t, \log \sigma^{(l)}_t)$ via a stack of residual blocks. We obtain a reduction in parameter count by sharing parameters across every 2 time-steps via 3-D convolutions in our residual blocks.

As shown in the right of Figure 10, each 3-D residual block consists of three layers. The first layer has a filter size of 2x3x3 with 512 output channels followed by a ReLU activation. The second layer has two $1 \times 1 \times 1$ convolutions via the Gated Activation Unit Van Den Oord et al. (2016); van den Oord et al. (2016a). The third layer has a filter size of $2 \times 3 \times 3$ with the number of output channels determined by the level. This block is replicated three times in parallel, with dilation rates 1, 2 and 4, after which the results of each block, in addition to the input of the residual block, are summed.

The first two layers are initialized using a Gaussian distribution and the last layer is initialized to zeroes. In that way, the residual network behaves as an identity network during initialization allowing stable optimization. After applying a sequence of residual blocks, we use the last temporal activation that should capture all context. We apply a final $1 \times 1$ convolution to this activation to obtain $(\Delta \mathbf{z}^{(l)}_t, \log \sigma^{(l)}_t)$. We then add $\Delta \mathbf{z}^{(l)}_t$ to $\mathbf{z}^{(l)}_{t-1}$ to a *temporal skip connection* to output $\boldsymbol{\mu}^{(l)}_t$. This way, the network learns to predict the change in latent variables for a given level. We have provided visualizations of the network architecture in this website

## E  ABLATION STUDIES

Through an ablation study, we experimentally evaluate the importance of the following components of our VideoFlow model: (1) the use of temporal skip connections, (2) the use Gated Activation Unit (GATU) instead of ReLUs in the residual network and (3) the use of dilations in $NN_{\boldsymbol{\theta}}()$ in Section D

We start with a VideoFlow model with 256 channels in the coupling layer, 16 steps of flow and remove the components mentioned above to create our baseline. We use four different combinations of our components (described in Fig. 11) and keep the rest of the hyperparameters fixed across those combinations. For each combination we plot the mean bits-per-pixel on the holdout BAIR-action

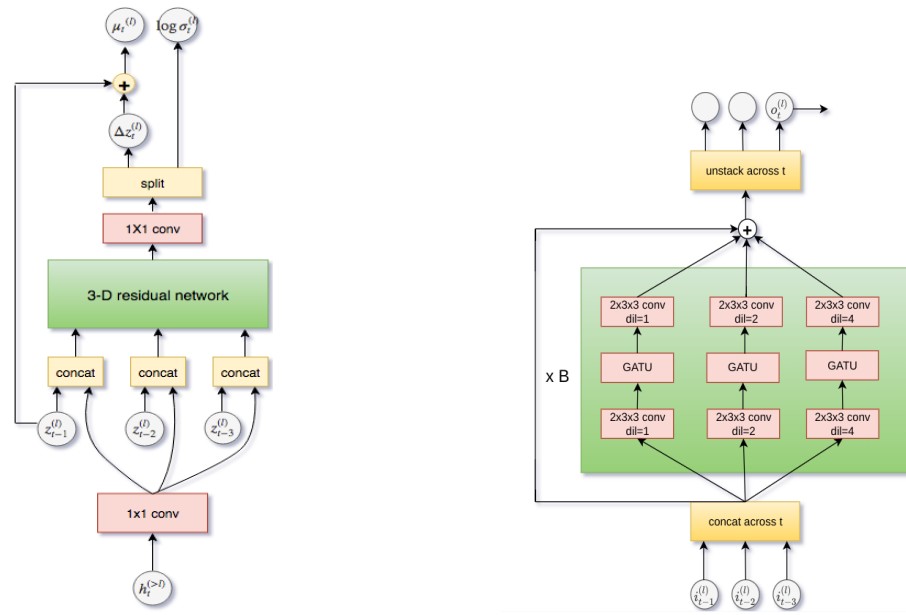

**Figure 10: Left:** We predict a gaussian distribution over $\mathbf{z}_t^{(l)}$ via a 3-D Residual network conditioned on $\mathbf{z}_{<t}^{(l)}$ and $\mathbf{z}_t^{(>l)}$. **Right:** Our 3-D residual network architecture is augmented with dilations and gated activation units improving performance.

free dataset over 300K training steps for both affine and additive coupling in Figure 11. For both the coupling layers, we observe that the VideoFlow model with all the components provide a significant boost in bits-per-pixel over our baseline.

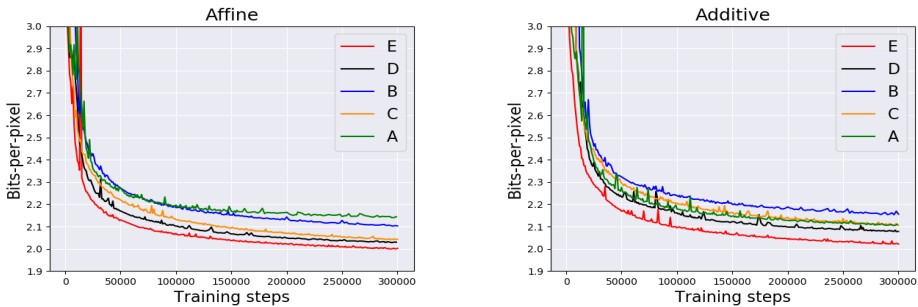

**Figure 11: B: baseline**, **A: Temporal Skip Connection**, **C: Dilated Convolutions + GATU**, **D: Dilation Convolutions + Temporal Skip Connection**, **E: Dilation Convolutions + Temporal Skip Connection + GATU**. We plot the holdout bits-per-pixel on the BAIR action-free dataset for different ablations of our VideoFlow model.

We also note that other combinations—dilated convolutions + GATU (C) and dilated convolutions + the temporal skip connection —improve over the baseline. Finally, we experienced that increasing the receptive field in $NN_{\boldsymbol{\theta}}()$ using dilated convolutions alone in the absence of the temporal skip connection or the GATU makes training highly unstable.

## F EFFECT OF TEMPERATURE ON SAVP-VAE AND SV2P

We repeat our evaluations described in Figure 4 applying low temperature to the latent gaussian priors of SV2P and SAVP-VAE. We empirically find that decreasing temperature from 1.0 to 0.0 monotonically decreases the performance of the VAE models. Our insight is that the VideoFlow

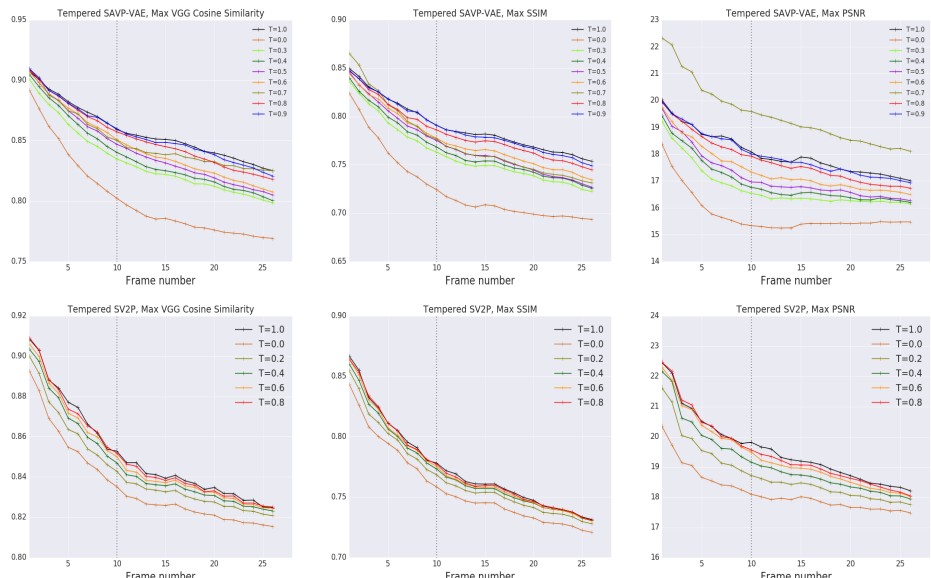

**Figure 12:** We repeat our evaluations described on the SV2P and SAVP-VAE model in Figure 4 using temperatures from 0.0 to 1.0 while sampling from the latent gaussian prior.

model gains by low-temperature sampling due to the following reason. At lower T, we obtain a tradeoff between a performance gain by noise removal from the background and a performance hit due to reduced stochasticity of the robot arm. On the other hand, the VAE models have a clear but slightly blurry background throughout from $T = 1.0$ to $T = 0.0$. Reducing T in this case, solely reduces the stochasticity of the arm motion thus hurting performance.

## G LIKELIHOOD VS QUALITY

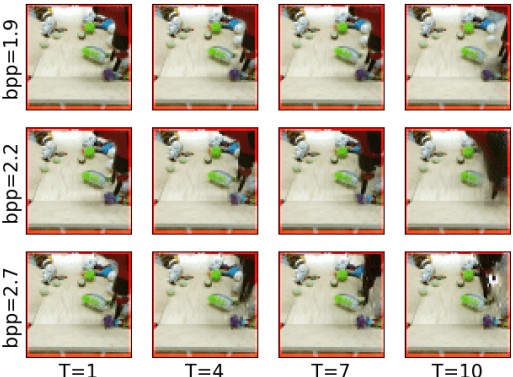

**Figure 13:** We provide a comparison between training progression (measured in the mean bits-per-pixel objective on the test-set) and the quality of generated videos.

We show correlation between training progression (measured in bits per pixel) and quality of the generated videos in Figure 13. We display the videos generated by conditioning on frames from the test set for three different values of bits-per-pixel on the test-set. As we approach lower bits-per-pixel, our VideoFlow model learns to model the structure of the arm with high quality as well as its motion resulting in high quality video.

## H    VideoFlow - BAIR Hyperparameters

### H.1    Quantitative - Bits-per-pixel

To report bits-per-pixel we use the following set of hyperparameters. We use a learning rate schedule of linear warmup for the first 10000 steps and apply a linear-decay schedule for the last 150000 steps.

| Hyperparameter | Value |
|---|---|
| Flow levels | 3 |
| Flow steps per level | 24 |
| Coupling | Affine |
| Number of coupling layer channels | 512 |
| Optimier | Adam |
| Batch size | 40 |
| Learning rate | 3e-4 |
| Number of 3-D residual blocks | 5 |
| Number of 3-D residual channels | 256 |
| Training steps | 600K |

### H.2    Qualitative Experiments

For all qualitative experiments and quantitative comparisons with the baselines, we used the following sets of hyperparameters.

| Hyperparameter | Value |
|---|---|
| Flow levels | 3 |
| Flow steps per level | 24 |
| Coupling | Additive |
| Number of coupling layer channels | 392 |
| Optimier | Adam |
| Batch size | 40 |
| Learning rate | 3e-4 |
| Number of 3-D residual blocks | 5 |
| Number of 3-D residual channels | 256 |
| Training steps | 500K |

## I    Hyperparameter grid for the baseline video models.

We train all our baseline models for 300K steps using the Adam optimizer. Our models were tuned using the maximum VGG cosine similarity metric with the ground-truth across 100 decodes.

**SAVP-VAE and SV2P:** We use three values of latent loss multiplier 1e-3, 1e-4 and 1e-5. For the SAVP-VAE model, we additionally apply linear decay on the learning rate for the last 100K steps. **SAVP-GAN:** We tune the gan loss multiplier and the learning rate on a logscale from 1e-2 to 1e-4 and 1e-3 to 1e-5 respectively.

## J    Correlation between VGG perceptual similarity and bits-per-pixel

We plot correlation between cosine similarity using a pretrained VGG network and bits-per-pixel using our trained VideoFlow model. We compare $\mathcal{P}(X_4 = X_t|X_{<4})$ as done in Section 5.5 and the VGG cosine similarity between $X_4$ and $X_t$ for $t = 4 \ldots 13$. We report our results for every video in the test set in Figure 15. We notice a weak correlation between VGG perceptual metrics and bits-per-pixel with a correlation factor of $-0.51$.

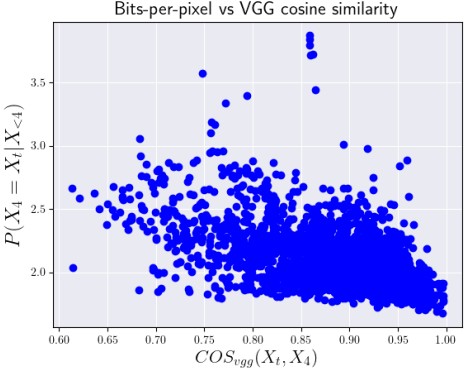

**Figure 14:** We compare $\mathcal{P}(X_4 = X_t | X_{<4})$ and VGG cosine similarity between $X_4$ and $X_t$ for $t = 4 \ldots 13$

## K  VIDEOFLOW: LOW PARAMETER REGIME

We repeated our evaluations described in Figure 4, with a smaller version of our VideoFlow model with 4x parameter reduction. Our model remains competetive with SVG-LP on the VGG perceptual metrics.

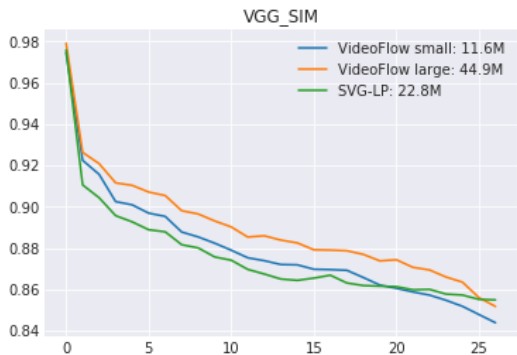

**Figure 15:** We repeat our evaluations described in Figure 4 with a smaller version of our VideoFlow model.

