# OpenReview forum: "VideoFlow: A Conditional Flow-Based Model for Stochastic Video Generation"
_ICLR.cc/2020/Conference — Accept (Poster)_

### Official Review · AnonReviewer3 · 2019-10-21
**Official Blind Review #3**

**Rating:** 6

**Review:**

This paper presents a stochastic model based on Glow for conditional video generation. The major novelty of this work is to introduce the flow-based models to video modeling and learn the video dynamics via the dependencies of the latent variables. The general idea is reasonable and the proposed model is technically correct, but I have the following concerns mainly about the originality and the experiments.

**Above all, most of the text in Section 3 is very similar (or exactly the same) to the text of the Glow paper (the background section).**

— Significance and originality —
a.1) In Section 1, the authors discussed some possible application scenarios of video prediction models, e.g. learning from unlabeled data and being used for downstream tasks. However, all models mentioned here, including [Mathieu et al. 2016] and [Finn et al. 2016], are deterministic models. Thus, in what way can the stochastic model proposed in this paper be used in real applications?

a.2) VideoFlow can be viewed as an extension of the Glow model. There are two problems. First, the originality is limited. I don’t think modeling the temporal dependencies of the latent variables with a convolutional network is a significant contribution to the conditional flow-based methods. Second, this paper is not self-contained. After reading Section 4.2, I have to check the previous literature to find the objective function, the network details, or the training procedure.

— Experiments —
b.1) Throughout the experiments, the VideoFlow model is mainly compared with two stochastic video prediction models that were probably proposed by the same research group. If it is possible, the authors might include other stochastic models such as the SVG-LP [Denton & Fergus 2018], and at least one deterministic model such as the E3D-LSTM [Wang et al. 2019] as well.
[Wang et al. 2019] E3D-LSTM: A Model for Video Prediction and Beyond.

b.2) The evaluation metric bits-per-pixel was not directly optimized by the previous video generation/prediction models. Thus, the comparisons in Table 2 might be unfair.

b.3) Since training the Glow model requires a huge computational cost, how is the training efficiency of the VideoFlow model compared with other stochastic video generation models?

— Other —
c.1) In Section 4, it is not clear what the temporal border effect means?

AFTER REBUTTAL:
Though the overall novelty is still not fully convincing, this paper may shed some insights into video generation by introducing flow-based models to this topic. I have increased my score from 3 to 6.

**Experience Assessment:**

I have published one or two papers in this area.

**Review Assessment: Checking Correctness Of Derivations And Theory:**

I assessed the sensibility of the derivations and theory.

**Review Assessment: Checking Correctness Of Experiments:**

I carefully checked the experiments.

**Review Assessment: Thoroughness In Paper Reading:**

I read the paper at least twice and used my best judgement in assessing the paper.

---

> ### Author Response · Authors · 2019-11-14
> **Response to AnonReviewer3 [1/2]**
>
> We thank you for your reviews. We believe we have addressed your concern about the writing in the revision. Please find attached our detailed response below.
>
> Experiments
> -----------------
> b1) SVG-LP: As requested, we added results from SVG-LP [Denton & Fergus 2018], another strong baseline to Figure 4 and Figure 3 in our revision. VideoFlow either outperforms or is competent with SVG-LP on all the metrics used in the paper.
>
> Deterministic baseline: We evaluated two deterministic baselines CDNA [Finn et al, 2016] and EPVA [Wichers et al, 2017] on the BAIR Robot pushing dataset. The samples from the CDNA model were qualitatively better as compared to the samples from the EPVA model. So, we added results from [Finn et al, 2016] on multiple metrics to Figure 4 of our revision.
>
> b2) We believe we made a good-faith effort; We estimated the best-possible beta for the video-VAE’s post training.  We also employed importance sampling using 100 samples from the posterior which gives a much tighter bound on the bits-per-pixel. Our goal was to measure how the VAE models perform out-of-the-box on a density estimation task.  In other words, in addition to being better / competent with the VAE approaches, our model also has the additional advantage of good likelihood numbers.
>
> b3) Our VideoFlow model reported in the paper has 45M parameters as compared to SVG-LP that has around 23M parameters. But, we performed the following experiments to make a convincing case.
>
> * We trained a smaller version of our model (VideoFlow small) with 12M parameters. We report our results in Section I of the appendix (VideoFlow: low parameter regime) in the revision. We are slightly better than SVG-LP on VGG perceptual metrics despite being 2x smaller.  We lose ~0.2 bpp as compared to VideoFlow large but our samples are still largely coherent. [2,3]
>
> * For VideoFlow small and VideoFlow large, we reported our results after 5 days and 2 weeks (600K steps) on 8 GPU’s respectively. But we gain very little (around 0.04-0.05 bpp) between training our model for 200K and 600K steps.  We attached our bits-per-pixel on the validation set as a function of training steps over here which validates this claim [1]. We were able to generate high quality samples within 150-200K steps. We expect our results should be comparable or slightly worse if at all, when evaluated on a checkpoint at 200K steps. In comparison, the video-VAE models were trained between 2-3 days on 1 GPU.
>
> In future, we could leverage improvements in normalizing flows to further close this gap.
>
> Significance
> ----------------
> a.1) Good examples are [Hafner et al 2018] and [Kaiser at al 2018] where they report higher scores in numerous planning and reinforcement learning tasks by utilizing stochastic video prediction models. Also, [Nair et al 2018], leverage a stochastic video prediction model for self-supervision.
>
> In short, a deterministic model cannot accurately model settings where there are multiple possible outcomes, (most real-life settings) and is obliged to predict a statistic of all the possible outcomes.
>
> Other
> --------
> We replaced this with “without introducing such artifacts” in our latest version.
>
>
> [1] https://ibb.co/NNrwfGm
> [2] https://gifyu.com/image/v9Rp
> [3]  https://gifyu.com/image/v9RT

---

> > ### Author Response · Authors · 2019-11-14
> > **Response to AnonReviewer3 [2/2]: Originality**
> >
> > Originality
> > ---------------
> >
> > a.2)
> >
> > Changes made to the writing
> > ----------------------------------------
> >
> > We agree that some parts of the paper assumes background knowledge of the multi-scale architecture used. We have made the following changes to our paper to alleviate that concern.
> >
> > * We added Section 4.1 “Invertible Multi-Scale Architecture” to our revision. In our new section, we briefly explain the multi-scale architecture. We describe the invertible transformations used in the multi-scale architecture and how the per-frame latent variables per level (scale) are inferred to make it more self-contained.
> >
> > * We added the second last paragraph under Section 4.2 where we explain how the invertible multi-scale architecture and the autoregressive latent dynamics model contribute to different parts of the objective.
> >
> > * We restructured Section 3, to describe the training objective.
> >
> > Technical contributions
> > ---------------------------------
> >
> > We believe we have made the following technical contributions via our paper:
> >
> > * A stochastic invertible flow-based model for video that is able to compute exact likelihoods, exact latent inference and fast sampling (as compared to autoregressive models). In addition to the above advantages, we provide extensive evaluations demonstrating that VideoFlow is either comparable or outperforms very strong baselines on a number of metrics. We qualitatively show latent-space interpolations demonstrating that VideoFlow learns meaningful latent representations [1] (Section 5.3).
> >
> > * We describe the first approach that scales the normalizing flow technique to video. Our model generates very high quality samples encouraging further investigation of flow-based models in the video generation literature.
> >
> > * We augmented a standard convolutional network with modifications in Section B of the appendix; we ablate these modifications demonstrating performance improvement in Section C of the appendix. We added a couple of lines at the end of Section 4.2 and provide a network diagram (Figure 8 in the appendix) to make our architecture clear.
> >
> > * We provide well documented code for other researchers to build on top of our work.
> >
> > [1] https://sites.google.com/view/videoflow/home#h.p_T96KYHZ0jtCQ

---

### Official Review · AnonReviewer1 · 2019-10-23
**Official Blind Review #1**

**Rating:** 6

**Review:**

The paper "VideoFlow: A Conditional Flow-Based Model for Stochastic Video " proposes a new model for video prediction from a starting sequence of conditionning frames. It is based on a state-space model that encodes successive frames in a continuous hierarchical state, with contraints on trajectories of the codes in this state.

I like the invertible NN framework the model relies on. It allows to avoid variational autoencoding of frames via invertible deterministic transforms. Learning the dynamics of the video is therefore easier, since there is no need of any stochastic inference process.    However, is there no risk of high latent vacancy in the representation space? Uncertainty of stochastic inference usually helps filling the space by considering larger areas of codes than deterministic process. Also, since at each step, the next code is conditionned by the whole past sequence of codes, besides the increasing complexity induced, I am wondering if such a model is able to efficiently encode the dynamics and the stochasticity of the video. In fact, a given z_t does not encode any dynamics nor uncertainty at that point, only the image (it cannot since it is fully determined via the invertible function from the image). Imagine that at a given point, two very different scenarios can follow, with very different following frames. In that case, how could the next state could encode these two different futures with a simple gaussian in the space ? Also,  it would be useful to compare the model with a version where the invertible frame encoder and the sequential model would be learned separately, to better understand what the model really does during training. A study of the impact of the hierarchy depth would also be useful.

Also, an additional real-world dataset would be useful for really assessing the performance of the model, since BAIR is known to be fully random and the past does not highly impact the future. A possible dataset would be KTH. Other baselines could also be considered, notably the famous approach from  [Denton et al., 2017].

At last, the clarity of some parts could be improved. Notably the description of the sequential model in the space, whih is succintly given in the appendix.



**Experience Assessment:**

I have published one or two papers in this area.

**Review Assessment: Checking Correctness Of Derivations And Theory:**

I assessed the sensibility of the derivations and theory.

**Review Assessment: Checking Correctness Of Experiments:**

I assessed the sensibility of the experiments.

**Review Assessment: Thoroughness In Paper Reading:**

I read the paper at least twice and used my best judgement in assessing the paper.

---

> ### Public Comment · ~Jessie_Yuan1 · 2019-11-07
> **Human Motion Datasets**
>
> If I recall correctly, the authors' demo in a workshop showed that their model fails to capture reasonable human motions. It would really be interesting to know if they have improved on this.

---

> ### Author Response · Authors · 2019-11-14
> **Response to AnonReviewer [1/2]**
>
> We thank you for your reviews. Please find attached our response.
>
> Q: Clarity can be improved:
>
> We added the following changes to our revision.
> 1. We added network diagrams of the 3-D residual network used to model temporal dependencies (Figure 8) in the appendix, to assist the description of the sequential model in the appendix (Section B).
> 2. We added Section 4.1, where we briefly explain the multi-scale architecture before moving on to the autoregressive latent dynamics model. We describe the invertible transformations used in the multi-scale architecture and how the per-frame latent variables per level (scale) are inferred.
> 3. We added the second last paragraph under Section 4.2 that describes how the invertible multi-scale architecture and the autoregressive latent dynamics model contribute to different parts of the training objective.
>
> Q: Additional baseline and dataset
>
> As requested, we added results from SVG-LP [Denton & Fergus 2018], another strong baseline to Figure 4 and Figure 3. VideoFlow either outperforms or is comparable to SVG-LP on all metrics in the paper. We also added results from CDNA [Finn et al. 2016], a strong deterministic baseline to our results in Figure 4.
>
> In regard to our choice of the BAIR dataset for comparisons, it is a standard evaluation benchmark used in the stochastic video prediction literature. We believe the BAIR robot dataset is challenging due to its stochasticity i.e. there are multiple possible futures for the robot arm in the absence of supervision via actions as well as unknown physical properties of the objects. We agree that including experiments on larger and high resolution datasets would indeed be even more interesting to explore in future
>
> Q: Learning the invertible encoder and sequential model separately
>
> We did attempt training the invertible encoder and sequential model separately in our initial experiments. We compared:
> 1. Training the sequential model and the invertible flow encoder jointly. (our current version)
> 2. Two stage training process:
>       Stage a): Pretraining the invertible flow encoder to model individual frames that provides stable latent representations.
>       Stage b): (Training the sequential model + Fine-tuning the flow encoder) on video.
>
> We found out that after pre-training the invertible flow encoder (i.e 2a), (2b) does indeed converge faster as compared to 1.
> But the total compute time of 2 (2a + 2b), was similar to (1). In addition this training scheme added increased complexity to our model, so we disbanded this after our initial efforts.
>
> Q: Impact of hierarchy depth
>
> With a flow level of 1, our generated samples were able to capture the global structure of the robotic arm (for eg, a red blob).  This is similar to Fig 9 in [Kingma & Dhariwal, 2018], where a flow model with a lower number of levels of hierarchy captures global structure. We also show qualitatively in [1] and Section 5.3, that the latents at lower levels encode background objects as smaller scales while higher levels encode larger objects, such as the robotic arm.
>
> Q: .... The next code is conditioned by the whole past sequence of codes, besides the increasing complexity induced.....
>
> For computational efficiency, we limit the history of the codes that we condition on to a window of 3 frames. We report this in Section 5.2 and the first paragraphs of Section 5.4. We empirically find that this sufficient to infer the dynamics of the dataset. This works quite well, but we do see that this Markovian assumption does have artifacts in the case of occlusions and obstructions that we report in Section 5.4 (Longer predictions)

---

> > ### Author Response · Authors · 2019-11-14
> > **Response to AnonReviewer [2/2]**
> >
> > Q: If such a model is able to efficiently encode the dynamics and the stochasticity of the video. In fact, a given z_t does not encode any dynamics nor uncertainty at that point, only the image.
> >
> > Our hypothesis agrees with your insight. We hypothesize that the flow model by itself encodes the frame into meaningful latent codes that are state (frame) specific while the sequential model learns the dynamics of the video, i.e how the latent codes evolve over time. We provide additional insight using latent space interpolations between the first and last frame using the trained invertible flow encoder here [1]. We observe that the robotic arm and objects move in a smooth trajectory in between the first and last position. This indicates that the flow encoder encodes useful state-specific information such as the position of the robotic arm and background objects as latent codes.
> >
> > Q: Imagine that at a given point, two very different scenarios can follow, with very different following frames. In that case, how could the next state could encode these two different futures with a simple gaussian in the space ?
> >
> > Let us say that the flow encoder encodes meaningful information such as the position of the arm as latent codes. In that case, we can effectively model, the position of the robot arm at time T as a gaussian, whose mean and variance are parametrized by a network conditioned on the previous few positions. By doing this, we allow the sequential model to infer the dynamics of the video for e.g, velocity. The invertible flow model can then generate a high-quality frame using the latent code at time T.
> >
> > We empirically show that this uncertainty in latent space as given by a gaussian prior translates to highly diverse trajectories in pixel space over multiple time-steps both qualitatively [2] and quantitatively (Figure 3 in our paper)
> >
> > We can also control this uncertainty in latent space using temperature; indeed we show that when we reduce uncertainty / diversity in latent space, relates to reduced stochasticity and diversity in trajectories in pixel space [3].
> >
> > [1] https://sites.google.com/view/videoflow/home#h.p_T96KYHZ0jtCQ
> > [2] https://sites.google.com/view/videoflow/home#h.p_qrclMoIvHzNC
> > [3] https://sites.google.com/view/videoflow/home#h.p_NCseKczbThPX

---

### Official Review · AnonReviewer2 · 2019-10-23
**Official Blind Review #2**

**Rating:** 6

**Review:**

This paper extended the flow-based generative model for stochastic video prediction. The proposed model takes an advantage of the flow-based models which provide exact latent-variable inference, exact log-likelihood evaluation, and efficiency. The paper used the autoregressive model and the multi-scale Glow architecture. The experiments on the stochastic movement dataset (synthetic) and the BAIR Robot push dataset show the performance improvement against other state-of-the-art stochastic video generation models (SV2P and SAVP-VAE).

The main contribution in this paper is the use of flow-based models for video prediction, and it is the first work in this direction. The major idea sounds and the paper is clearly written.

Below is my concerns and the feedback.

It looks like the low-temperature sampling is important to achieve the better scores for prediction. Can the low-temperature sampling trick be applied for SV2P and SAVP-VAE as well? If then, how is the performance difference compare to the proposed model?

The authors reported the best possible values of PSNR, SSIM and VGG perceptual metrics by choosing the video closest to the ground-truth. However, I believe this evaluation does not present the benefit of the stochastic models. The better comparison I believe is to report the median/mean with the range between best and worst values.

The BAIR robot push dataset is with a pretty limited setting: a small robot and/or object motion between frames and a small variation of the background between videos. It would be interesting to see more dynamic scenarios such as driving or human motion scenes.

**Experience Assessment:**

I have published in this field for several years.

**Review Assessment: Checking Correctness Of Derivations And Theory:**

I assessed the sensibility of the derivations and theory.

**Review Assessment: Checking Correctness Of Experiments:**

I carefully checked the experiments.

**Review Assessment: Thoroughness In Paper Reading:**

I read the paper thoroughly.

---

> ### Author Response · Authors · 2019-11-14
> **Response to AnonReviewer2**
>
> We thank for your reviews. Please find attached our response.
>
> Low temperature Sampling
> --------------------------------------
> As suggested, in the updated version of the paper, we included experiments in which we applied low-temperature sampling to the latent gaussian priors of SV2P and SAVP-VAE. We report our results in Section D (Effect of Temperature on SAVP-VAE and SV2P ) of the appendix in the revision. We empirically find that decreasing temperature from 1.0 to 0.0 monotonically decreases the performance of the VAE models.
>
> Our insight is that the VideoFlow model gains by low-temperature sampling (upto a certain temperature) due to the following reason. By decreasing the temperature of the flow model, we trade-off between a performance gain by noise removal from the background and a performance hit due to reduced stochasticity of the robot arm.
>
> On the other hand, the VAE models have a clear but slightly blurry background throughout from T=1.0 to T=0.0. Reducing T in this case, solely reduces the stochasticity of the arm motion thus hurting performance.
>
> Reporting best vs mean
> ---------------------------------
> In the updated version paper, we added a summary of our response below to the sub-section "Accuracy of the best sample" in Section 5.2 to make this clear.
>
> The BAIR dataset is highly stochastic and the number of plausible futures are high. Each generated video can be super realistic, can represent a plausible future in theory but can be far from the single ground truth video perceptually. The best values according to the PSNR, SSIM and VGG metrics from a finite number of samples is a proxy to help us understand if the ground truth can lie in the set of possible futures as per the model.
>
> Consider a hypothetical scenario where there are eight plausible but completely diverse future frames, such that the pairwise perceptual similarity between the frames ~= 0.0. Let us also consider the perfect model that is capable of generating each of these future frames accurately.
>
> If we, compute the similarity of each sample with the ground truth video, and average this across multiple samples, we would get a similarity of ~ 12.5%. The “mean” in this case is not a useful statistic and the “best” quantifies the performance of the stochastic model better. This metric should be also used in combination with the Amazon MTurk results in Figure 3 and Section 5.2 to assess if the generated videos are realistic.
>
> BAIR Robot dataset
> ----------------------------
> In regard to our choice of the BAIR dataset for comparisons, it is a standard evaluation benchmark used in the stochastic video prediction literature. [Babazeidah et al 2018, Lee et al 2018, Unterthiner et al. 2018, Denton & Fergus 2018, Weissenborn et al. 2019]. We believe the BAIR robot dataset is challenging due to its stochasticity i.e. there are multiple possible futures for the robot arm in the absence of supervision via actions as well as unknown physical properties of the objects. A network unable to model the stochasticity (e.g. a deterministic network) would blur the arm out in all possible directions. We agree that including experiments on larger and high resolution datasets would indeed be even more interesting to explore in future.

---

### Author Response · Authors · 2019-11-14
**Rebuttal**

Thanks everyone for the time and reviews. Here is a summary of changes made to the draft.

Experiments
------------------
1. Added SVG-LP baseline to Figure 3 and Figure 4.
2. Added low-temperature sampling results on the video-VAE models to Section D of the Appendix.
3. Added CDNA, a deterministic video prediction baseline to Figure 4.
4. Added comparison with SVG-LP on VGG perceptual metrics in the low-parameter regime to Section I of the appendix.

Writing
--------
1. Added Section 4.1 “Invertible multi-scale architecture” that briefly summarizes the invertible flow architecture.
2. Added Figure 8, network diagrams to assist the description of the residual network architecture in Section C of the appendix.
3. Added a bit on the training objective (second last paragraph) of Section 4.2.
4. Restructured Section 3.

---

### Public Comment · ~Jianwen_Xie1 · 2020-01-02
**Some related work about different classes of video generative models (energy-based, CoopNets-based, ABPTT-based)**


Dear Authors,

Congratulations on your accepted paper.

I would like to share you some papers that are related to different classes of generative models for stochastic video generation.

Besides GAN-based, VAE-based, and Flow-Based video generative model, there are (1) deep energy-based, (2) CoopNet-based, and (3) ABPTT-based models.  All (1)(2)(3) are likelihood models.

(1) deep energy-based video generative model (Xie. CVPR 2017; Xie. PAMI 2019) (It uses a spatial-temporal ConvNet to parameterize the energy function and learns the model via MCMC-based Maximum likelihood; It is a single model without using extra assisting network. The generation is based on Langevin dynamics. )

Synthesizing Dynamic Pattern by Spatial-Temporal Generative ConvNet
Jianwen Xie, Song-Chun Zhu, Ying Nian Wu (CVPR 2017)

Learning Energy-based Spatial-Temporal Generative ConvNets for Dynamic Patterns
Jianwen Xie, Song-Chun Zhu, Ying Nian Wu (PAMI 2019)


(2) Generative Cooperative Nets (CoopNets), which cooperatively trains deep EBM (Xie. CVPR 2017; Xie. PAMI 2019) and the generator by MCMC teaching.  The whole framework are based on MLE. This is a two-model framework, different from VAE and GAN. The paper shows video generation by CoopNets in the last experiment in (Xie PAMI 2018).

Cooperative Training of Descriptor and Generator Networks
Jianwen Xie, Yang Lu, Ruiqi Gao, Song-Chun Zhu, Ying Nian Wu
IEEE Transactions on Pattern Analysis and Machine Intelligence (TPAMI) 2018

(3) Dynamic generator (Xie. AAAI 2019) as a deep latent space model or deep auto-regressive model. The model is trained by MLE with an Alternating Back-Propagation Through Time (ABPTT) algorithm, where true Bayesian inference is performed via Langevin dynamics.  It is a single-model framework without using any extra net for adversarial or variational training.

Learning Dynamic Generator Model by Alternating Back-Propagation Through Time
Jianwen Xie *, Ruiqi Gao *, Zilong Zheng, Song-Chun Zhu, Ying Nian Wu (* equal contributions)
The Thirty-Third AAAI Conference on Artificial Intelligence (AAAI) 2019

I believe these papers regarding different classes of video generative models will make the related work of your paper more complete.

Thank you !

---

### Decision · Program_Chairs · 2019-12-19

**Decision:**

Accept (Poster)

**Comment:**

The authors explore the use of flow-based models for video prediction. The idea is interesting. The paper is well-written. It is a good paper worthwhile presenting in ICLR.

For final version, we suggest that the authors can significantly improve the experiments: (1) report results on human motion datasets; (2) include the results by the FVD metric.